# Unsupervised membrane subtraction in cryogenic electron microscopy images

## Abstract

Cryogenic electron microscopy (cryo-EM) of membrane proteins often requires extracting them from their membrane to simplify downstream image processing. While this step reduces the influence of membranes on 3D reconstruction, it also prevents proteins from being observed in their natural state. To overcome this limitation, we propose a two-step machine learning framework that avoids protein extraction: (1) membrane detection, which identifies the bilayer membrane, and (2) membrane subtraction, which digitally removes the detected membrane from the cryo-EM micrograph. Recent work has introduced supervised algorithms for membrane detection, but membrane subtraction remains relatively underexplored. Here, we present a novel unsupervised approach to membrane subtraction that models membranes using a general representation and computes a smooth estimate, which can then be subtracted from the original cryo-EM micrograph. Experimental results show that our method outperforms existing membrane subtraction alternatives and enables reliable 3D reconstruction of membrane proteins using cryo-EM without protein extraction.

## 1 Introduction

In structural biology, cryogenic electron microscopy (cryo-EM) is a popular structure determination technique for identifying the three-dimensional (3D) atomic structure of isolated protein complexes (i.e., single particle analysis [SPA] (Henderson, 2017)). Cryo-EM has led to a rapid increase in the number of proteins with solved structures (PDB).

A significant challenge in cryo-EM (and structural biology) is the determination of membrane protein structures and understanding the relation of their structures and functional states (e.g., open or closed states of an ion channel). Membrane proteins have hydrophobic regions that traverse the lipid bilayer membrane of a cell or a vesicle, and imaging membrane proteins while they are inserted in the membrane is critical to understanding their native structure. Consequently, in SPA experimental practice, membrane proteins are often isolated from their membranes before imaging. However, membrane proteins can be imaged in their biologically native cellular or vesicle membrane, or imaged after insertion into a synthetic vesicle membrane. Figure 1(a)-(c) show portions of SPA cryo-EM micrographs containing different vesicles and their embedded membrane protein.

A cryo-EM micrograph (or image) can be thought of as a tomographic projection of 3D objects in the specimen followed by the action of the point spread function of the electron microscope (Frank, 2006). When membrane proteins are imaged while embedded in a bilayer membrane, the bilayer membrane ($\sim 50\text{Å}$ in thickness) contributes a strong signal when oriented parallel to the electron beam. This contribution can be significantly larger than that of the protein embedded in the membrane (50-100Å in size). The strong membrane signal can cause SPA reconstruction to fail in the so-called "alignment phase" of the 3D reconstruction.

If the bilayer membrane can be detected and subtracted from the micrograph, then standard SPA reconstruction software can be used and the structure of the membrane protein elucidated. Since bilayer membranes are clearly visible in micrographs, they can be manually outlined to create a training dataset that will enable automated membrane detection by supervised learning. As discussed in Section 2 below, a number of software packages have recently been reported for automatic membrane detection. However, membrane subtraction remains a challenge for two reasons. First, membrane subtraction is not easily amenable to supervised learning. There is no manual technique

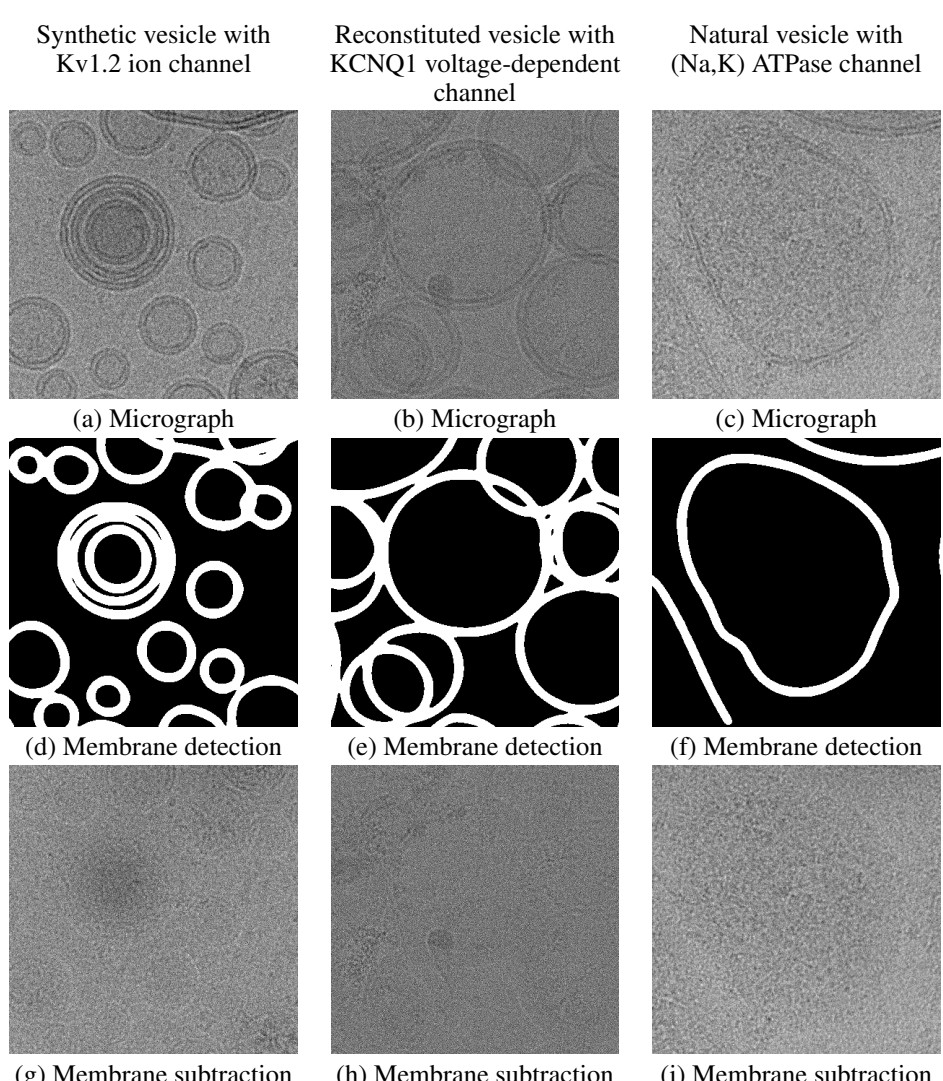

Figure 1: Portions of SPA cryo-EM micrographs of vesicles with embedded membrane proteins (a)-(c). Results of membrane detection (d)-(f) and membrane subtraction (g)-(i). See Section 4 for a more detailed description.

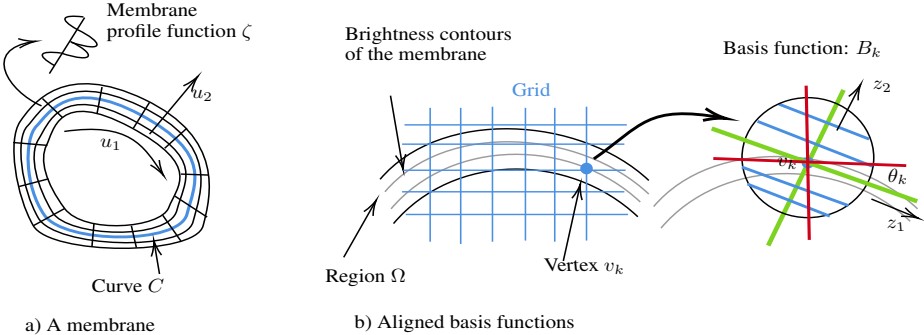

Figure 2: Membrane model and aligned basis functions.

of producing a mask of the micrograph with the membrane subtracted. Second, membrane subtraction is required to leave the protein intact. This is a complex problem since the structure of the protein, and hence its contribution to the image, is unknown.

The goal of this paper is to demonstrate an unsupervised approach to membrane subtraction in SPA cryo-EM micrographs. The key idea of our algorithm is to propose a general model for the appearance of the membrane in a micrograph. This model is used to create a set of basis functions which are capable of accurately representing the membrane. Projecting the image onto the span of these basis functions gives a smooth estimate of the membrane in the micrograph. This process can then be iterated – smooth estimates of the membrane can give even smoother basis functions, and projecting on these bases improves the membrane estimate. Subtracting this estimate from the original micrograph produces a membrane-subtracted micrograph. As we show below, the membrane estimate produced by our method surpasses the estimates produced by other methods, including deep learning-based denoising methods. In addition, the membrane subtracted images produced by our method resulted in membrane protein 3D reconstructions where the transmembrane domain of the protein is clearly visible.

**Innovation:** There are two key innovations in this paper: (1) the mathematical framework for representing and estimating membranes in an image. (2) the iterative unsupervised estimation of a membrane in a micrograph. As will become clear below, our definition of a membrane is quite general, and it and the membrane subtraction algorithm, can be used with other image features which appear membrane-like but are not a lipid bilayer membrane. The main innovation in our mathematical framework is a new method for estimating a profile of curved membrane when the membrane curve is unknown. The basic idea is described in the Proposition 1 in Chapter 3.

## 2 RELATED WORK

Multiple deep-learning algorithms for membrane detection in cryogenic electron tomography (cryo-ET) (Baumeister (1999); Lučić et al. (2005)) have recently been developed, including Mem-Brain Lamm et al. (2022) (here referred to as MemBrain v1), MemBrain v2 Lamm et al. (2025), and TARDIS Kiewisz et al. (2025). These membrane detectors are developed using a supervised learning paradigm with manually labeled 3D tomograms (not 2D micrographs). TARDIS has also been applied to SPA cryo-EM micrographs and is integrated into the CZI CryoET Data Portal Ermel et al. (2024) for automated membrane segmentation of cryo-ET datasets.

MemBrain v1 and v2 both use the U-Net architecture (Ronneberger et al., 2015) for tomogram segmentation, with MemBrain v2 further incorporating active learning to allow expert users to refine detector training. TARDIS uses an FNet architecture (a custom designed dual-decoder U-Net) combined with point cloud generation to achieve instance segmentation of membranes. According to the MemBrain authors' most recent preprint Lamm et al. (2025), segmentation accuracy for membrane detection is comparable between MemBrain v2 and TARDIS. However, TARDIS provides a more generalized framework that extends beyond membranes to additional cellular structures such as microtubules. To the best of our knowledge, none of these models currently perform membrane subtraction in either SPA cryo-EM micrographs or cryo-ET tomograms.

In a series of papers, Sigworth and colleagues Wang et al. (2006); Liu & Sigworth (2014); Jensen et al. (2016b;a) developed a semi-automatic algorithm for membrane detection and membrane subtraction of vesicles in micrographs. This algorithm is available at: (`https://github.com/fsigworth/aEMCodeRepository/tree/master/VesicleML`). Below we refer to this algorithm as the semi-automatic (SA) algorithm. Briefly, SA works as follows: An initial vesicle detection is carried out by correlating the micrograph with circular templates (2D projections of spherical membrane vesicles) and identifying local maxima. Then, the detected vesicles are chosen individually for further processing, which allows a more detailed modeling of the membrane appearance. However, both the detection step and the modeling step allow only small deviations from a circular shape, and model-fitting is very compute-intensive, and requires some manual interaction.

In principle, a possible solution to membrane subtraction is via image denoising, using either statistical methods such as Block-Matching and 3D-filtering (BM3D) (Mäkinen et al., 2020; Dabov et al., 2007) or deep-learning methods such as SwinIR (Liang et al., 2021). Potentially, denoising algorithms can filter out high-frequency signals, such as noise and proteins, leaving a bare image

of membranes. This denoised image can be subtracted from the micrograph. One limitation of these algorithms is that they are generic, i.e. they do not have an explicit understanding (model) of a lipid bilayer membrane and it is possible that the denoised estimate of the membrane is not membrane-like.

## 3 THEORY

### 3.1 WHAT IS A MEMBRANE?

Suppose $C$ is a smooth curve in a plane with a finite curvature (Figure 2(a)), called the *generating curve*. Let $\Omega$ be the set of points that are a fixed distance $\Delta$ away from $C$. Assume that $\Delta$ is small enough so that normals from any two points of the curve $C$ do not intersect within $\Omega$. Under that condition, there exists a curvilinear coordinate system in $\Omega$ with coordinates $u_1, u_2$ (see Figure 2(a)) such that $u_1 =$ constant are lines normal to the curve, and $u_2 =$ constant are curves parallel to $C$ (i.e. at a fixed distance from $C$). Given any point $x \in \Omega$, let $\chi$ be the "change of coordinate" function from the image coordinates to the curvilinear coordinates, i .e. $\chi(x) = u = (u_1, u_2)$, and $\chi^{-1}(u) = x$. A *membrane image*, or simply *a membrane*, is function $M : \Omega \to R$ defined by

$$M(x) = \zeta(\pi_2(\chi(x))), \tag{1}$$

where $x$ is a point in $\Omega$, $\pi_2((u_1, u_2)) = u_2$ is the projection on the second coordinate of $u = (u_1, u_2)$, and $\zeta$ is the *membrane profile function* (Figure 2(a)). The membrane profile function is essentially a normal "cross-section" of the membrane image. Typically, the membrane profile function is not a constant function. It demonstrates multiple local extrema. Of course, an image may contain one or more membranes.

This membrane model has a useful property: The contours of $M$ are curves corresponding to $u_2 =$const. Thus, the gradients of $M(x)$ in a small neighborhood of any $x \in \Omega$ either point along the $+u_2$ or $-u_2$ direction. This observation gives a simple criterion for evaluating how membrane-like an image $I$ is in a region $\Omega$. If $n(x) = \nabla I(x)/\|\nabla I(x)\|$ is the direction of the gradient, then $\partial_t(x) = n^T(x)H(x)n(x)$, where $H(x)$ is the Hessian of $I$ at $x$, is the component of the derivative of the gradient in the direction of the gradient. Similarly $\partial_p(x) = n_\perp^T(x)H(x)n(x)$, where $n_\perp(x)$ is $n(x)$ rotated by $90^o$, is the derivative of the gradient in the direction perpedicular to the gradient. For a membrane, $\partial_p(x)$ should be zero, or close to zero, whereas $\partial_t(x)$ should be significantly non-zero, reflecting the non-constant nature of the membrane profile function. Thus,

$$\nu = 1 - \frac{\text{rms}(\partial_p)}{\sqrt{\text{rms}^2(\partial_t) + \text{rms}^2(\partial_p)}}, \tag{2}$$

where rms is the root mean square value over $\Omega$, serves as an index of how membrane-like $I$ is in $\Omega$. This *membrane similarity index* takes values between 0 and 1, with higher values indicating that $I$ is more membrane-like in $\Omega$.

### 3.2 MEMBRANE ESTIMATION

Given a noisy image containing membranes, suppose that we have a membrane segmentation algorithm which segments the image into membrane and non-membrane regions. We take the membrane region to be $\Omega$, and hold it fixed. We do not require $\Omega$ to be the precise support of the membrane, only that $\Omega$ contain the membrane. Our task is to smooth the image content in $\Omega$ so that a good estimate of the membrane is obtained. This estimate is subtracted from the image to get the membrane-subtracted image.

Ideally, we would like to smooth the image in $\Omega$ along the $u_1$ coordinate of the membrane (Figure 2(a)). But this is difficult because we do not know the generating curve $C$. Instead we use the following strategy for smoothing: We impose a grid in $\Omega$ (Figure 2(b)) with $N$ vertices $v_k, k = 1, \cdots, N$. At each vertex $v_k$ we find a function $B_k$, called the *basis function at $v_k$*, which is a tangentially smoothened estimate of the local membrane profile near $v_k$ (Figure 2(b)). Projecting the image onto the span of the $B_k$'s gives a smooth version of the membrane, where the smoothing has the same spatial extent as that of the $B_k$'s (which is small). For increased smoothing, we re-estimate the $B_k$'s from the projected image, and re-project the image onto their span. This is carried

on iteratively till the image is sufficiently smoothed. Mathematically, the iteration is

$$\{B_k^n\} = \Phi(\hat{I}^n), \tag{3}$$

$$\hat{I}^{n+1} = \Pi_n(I), \ n = 1, \cdots \tag{4}$$

Equation (3) represents the creation of the set of basis functions $\{B_k^n\}$ from $\hat{I}^n$, the smoothened image at iteration $n$, by an operator $\Phi$. Equation (4) represents the projection of $I$, the original image (micrograph), onto the span of $\{B_k^n\}$. The iterations in Equations (3)-(4) are initialized with $\hat{I}^1 = I$. The details of the operator $\Phi$ and the projection $\Pi_n$ are as follows:

**The operator $\Phi$:** Suppose that at vertex $v_k$ we have an estimate of the direction normal to the membrane. Then, we choose a coordinate system at $v_k$ such that its x-axis is tangential to the membrane and y-axis is normal to the membrane (shown in green in Figure 2(b)). We call this the *aligned coordinate system*, and denote its coordinates as $z_1, z_2$. Let $\hat{\theta}_k$ be the angle between the image coordinate system and the aligned coordinate system (Figure 2(b)), and let $R_{\hat{\theta}_k}$ be the rotation matrix that converts a vector from the image coordinate system to the aligned coordinate system. Suppose $\pi_2$ is a function which projects a pair of coordinates onto the second coordinate i.e. $\pi_2([z_1, z_2]^T) = z_2$, then $\pi_2 R_{\hat{\theta}_k}(x - v_k)$ gives the second coordinate of $x \in \Omega$ in the aligned coordinate system. Suppose $\hat{\zeta}_k^n$ is a function that estimates the membrane profile at $v_k$, then the basis function $B_k^n$ is

$$B_k^n(x) = \hat{\zeta}_k^n(\pi_2 R_{\hat{\theta}_k}(x - v_k))e^{-\|x - v_k\|^2/2\sigma^2}, \tag{5}$$

where the Gaussian function $e^{-\|x - v_k\|^2/2\sigma^2}$ makes $B_k$ significant only in a neighborhood of $v_k$. The estimate of the membrane profile $\hat{\zeta}_k^n$ at the coordinate $z_2$ in the aligned coordinate system is obtained by the tangential weighted average:

$$\hat{\zeta}_k^n(z_2) = \int_{-L}^{L} w(z_1)\hat{I}^n(R_{-\hat{\theta}_k}([z_1, z_2]^T))dz_1, \tag{6}$$

were $w$ is a weighting function described shortly below. The function $B_k^n$ a smooth approximation to the membrane at local to $v_k$. The operator $\Phi$ in Equation (3) is the calculation of all $B_k^n$'s from $\hat{I}^n$ according to Equations (5)-(6).

**The projection $\Pi_n$:** The projection $I$ onto the span of $B_k^n$'s is obtained by minimizing the following objective function:

$$J = \int_\Omega (I(u) - \sum_k \alpha_k B_k^n(u))^2 du, \ \text{whose solution is} \tag{7}$$

$$\{\hat{\alpha}_k\} = \arg\min_{\{\alpha_k\}} J, \ \text{and} \ \hat{I}^{n+1}(x) = \sum_k \hat{\alpha}_k B_k^n(x). \tag{8}$$

In the above equations, $I$ is the image (micrograph), $\hat{\alpha}_k$ are the basis coefficients obtained by minimizing $J$, and $\hat{I}$ is the smoothed membrane image.

It remains to describe how we estimate $\hat{\theta}_k$ and what we use as the weight $w$ in Equation (6).

**Estimating $\hat{\theta}_k$:** In principle, we can estimate the $\hat{\theta}_k$'s by minimizing the objective function $J$ of Equation (7) with respect to the $\hat{\theta}_k$'s as well as the $\alpha_k$'s. However, this minimization is quite complex. Instead we use a heuristic for estimating $\hat{\theta}_k$, which works well in practice. The heuristic depends on the observation that $B_k^n$ is a good local approximation to the membrane at the correct angle $\hat{\theta}_k$. Thus, we estimate one $\theta_k$ at a time by:

$$\hat{\theta}_k = \arg\min_{\theta_k} \int_\Omega (\hat{I}^n(u) - B_k^n(u))^2 du. \tag{9}$$

**The weight $w$:** The weight $w$ influences how close $\hat{\zeta}_k^n$ approximates the profile $\zeta$ of $\hat{I}^n$ at $v_k$. The relation between the two is given by the following proposition (a detailed discussion of the propositions given below and their proofs can be found in the supplementary material). We suppress the superscripts below:

**Proposition 1:** For $\hat{\zeta}$ and $\zeta$ as defined above:

$$\hat{\zeta}(z_2) = \zeta(z_2) \int_{-L}^{L} w(z_1)dz_1 + \frac{1}{2}\frac{\zeta'(R+z_2)}{R+z_2}\int_{-L}^{L} w(z_1)z_1^2 dz_1 + O(L^5),$$ 

(10)

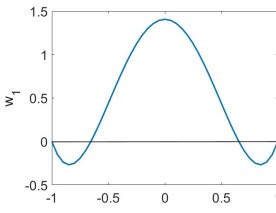

Figure 3: The optimal weight function $w_1(x)$.

where $R$ is radius of curvature of the membrane at $v_k$. Thus $\hat{\zeta}(z_2)$ can be a good approximation to $\zeta(z_2)$ for small $L$ if $\int_{-L}^{L} w(z_1)dz_1 = 1$ and $\int_{-L}^{L} w(z_1)z_1^2 dz_1 = 0$. Further, we require the weight $w$ to be smooth (to combat noise), i.e. to have a small value for $\int_{-L}^{L} w'^2(z_1)dz_1$. And we require $w$ to be even, and to go to zero at $\pm L$. A $w$ satisfying these criteria is available as:

**Proposition 2:** Amongst all $w \in C^2[-L, L]$ (the space of twice differentiable functions on $[-L, L]$), the even $w$ that minimizes $\int_{-L}^{L} w'^2(z_1)dz_1$ subject to $\int_{-L}^{L} w(z_1)dz_1 = 1$, $\int_{-L}^{L} w(z_1)z_1^2 dz_1 = 0$, $w(L) = w(-L) = 0$ is given by

$$w(z_1) = \frac{1}{L}w_1(\frac{z_1}{L}), \text{ where } w_1(x) = c_0 + c_2 x^2 + c_4 x^4,$$ 

(11)

and the coefficients $c_0, c_2, c_4$ satisfy

$$c_0 + c_2 + c_4 = 0, c_0 + \frac{1}{3}c_2 + \frac{1}{5}c_4 = \frac{1}{2}, \frac{1}{3}c_0 + \frac{1}{5}c_2 + \frac{1}{7}c_4 = 0.$$ 

(12)

Numerically solving Equation (12) gives $c_0 = 1.40625$, $c_2 = -4.6875$, and $c_4 = 3.28125$. The optimal weight function $w_1$ with these coefficients is shown in Figure 3.

To summarize, the entire algorithm is: **Iterative profile estimation and subtraction:**

0. *Input:* $I$ (micrograph), $\Omega$ (membrane region)

1. *Initialize*: Set $n = 1$, $\hat{I}^1$ = micrograph.

2. *Iterate:*

   (a) At each vertex in $\Omega$, estimate $\hat{\theta}_k$ via Equation (9), and the basis functions $B_k^n$ via Equation (5).

   (b) Estimate $\alpha_k$'s and $\hat{I}^{n+1}$ via Equations (7)-(8).

   (c) $n \leftarrow n + 1$.

Finally, subtract $\hat{I}^n$ from the micrograph.

## 4 EXPERIMENTS

**Data:** The data we use consists of micrographs for three membrane proteins: (1) the Kv1.2 ion channel (Long et al., 2005), (2) human KCNQ1 voltage-dependent ion channel (Mandala & MacKinnon, 2023), and (3) Membrane vesicles rich in the (Na,K) ATPase ion pump, derived from dog kidney outer medulla (III, 1982). Micrographs were obtained by request from the authors of the publications mentioned. Sample micrographs are shown in Figure 1(a)-(c).

The Kv1.2 ion channel preparation was created by removal of detergent from a mixture of detergent-solubilized synthetic lipids and purified Kv1.2 ion channel protein (Wu et al., 2025). The vesicle suspension was applied to a graphene substrate film and rapidly frozen for cryoEM analysis. For the KCNQ1 ion channel, the ion channel protein was purified, reconstituted in a similar fashion, and imaged suspended in vitreous ice (Mandala & MacKinnon, 2023). The (Na,K) ATPase preparation was obtained from homogenization of dog kidney tissue, followed by density-gradient enrichment, and sonication to reduce the vesicle size. In these vesicles the Na-K ATPase constitutes about 50% of the membrane protein. The vesicles were imaged in vitreous ice.

All micrographs were obtained with Titan Krios microscopes operating at 300 kV using Gatan K3 cameras producing micrographs of 5760x4092 pixels. The Kv1.2, KCNQ1, and Na-K ATPase micrographs have pixel sizes of 1.06, 1.09, and 0.825 Angstrom, respectively. All micrographs were downsampled by a factor of 4 to increase processing speed.

Two datasets were created from the micrographs. **Data set 1:** This dataset was used to train a membrane detection model using supervised learning (described below), and also to assess the performance of the membrane subtraction algorithm. This dataset contained 72 Kv1.2 micrographs, 61 KCNQ1 micrographs, and 48 ATPase micrographs. Vesicles in these micrographs were manually outlined. 5 and 10 micrographs from each membrane protein set were left out as validation and testing micrographs, and the remaining micrographs ($N = 136$) as training. **Data set 2:** This dataset contained a separate 5,000 Kv1.2. micrographs that the trained membrane detection model and the unsupervised membrane subtraction algorithm were applied to. After membrane subtraction, these micrographs were then used to reconstruct the structure of the Kv1.2 ion channel with a standard cryo-EM image processing pipeline.

**Membrane detection:** Inspired by existing cryo-ET membrane detectors (Lamm et al., 2022; 2025; Kiewisz et al., 2025), a U-Net model was trained to segment membranes in SPA cryo-EM micrographs (using Data Set 1 mentioned above). The default U-Net architecture implemented in PaddleSeg v2.4.2 (PaddlePaddle, 2019; Liu et al., 2021) (without pretrained weights) was trained to detect two target pixel classes: (1) background and (2) membrane. Training examples were generated by randomly cropping paired micrographs and membrane mask images into 256x256 pixel patches. Input patches were standardized by applying Gaussian smoothing (standard deviation of 24 pixels), subtracting the smoothed patch to zero-center pixel intensities, and scaling by division by the standard deviation of the resulting patch. These standardized patches underwent further data augmentation, including random horizontal and vertical flips, random brightness, contrast, and saturation distortions. Model training was conducted with a batch size of 4 over 500,000 iterations. The AdamW optimizer ($\beta1 = 0.9$, $\beta2 = 0.999$, weight decay= 0.01) was used with a polynomial learning rate decay schedule, starting at 0.005 and decaying to 0 with a power of 2.0. The objective function used during model training was Dice loss (Sudre et al., 2017). The model was trained on one NVIDIA Titan Xp GPU. The trained U-Net successfully segments membranes (Figure 1(d)-(f)). Segmentation performance was evaluated using the Dice coefficient. The model achieved a Dice score of 0.81 (training) and 0.79 (testing). Note that we regard membrane detection as a standard image pre-processing step, and do not make any claims of novelty for this approach. The novel part of our algorithm is membrane subtraction; its results are described below.

**Membrane subtraction:** The algorithm described at the end of Section 3 was implemented using PyTorch v2.7.1 Paszke et al. (2019) and Kornia v0.8.1 E. Riba & Bradski (2020). The algorithm parameters were set at follows: the grid spacing for $v_k$ was set to 4 pixels, the basis functions $B_k$ had a radius of 13 pixels. The optimizing coefficients $\alpha_k$ were calculated by gradient descent with a learning rate of 0.025 and 30 iterations. A total of 3 iterations (Step 2 in the algorithm) were sufficient for a good membrane estimate. Figure 1(g)-(i) show the results of subtracting the membrane from Figure 1(a)-(c).

**Other methods:** For comparison with our method, we used the SA membrane subtractor (`https://github.com/fsigworth/aEMCodeRepository/tree/master/VesicleML`), two denoising algorithms: statistical collaborative filtering algorithm BM3D (Mäkinen et al., 2020; Dabov et al., 2007), and the transformer-based deep learning algorithm SwinIR (Liang et al., 2021). For these algorithms, input image pixel intensities were normalized to $[0, 1]$ before the denoising. Then the denoised images within the $\Omega$ region were treated as membrane estimates. For the BM3D algorithm we used the constant-level noise standard deviation of 0.15. For SwinIR we used the models pre-trained to denoise images corrupted by zero mean and 25 and 50 std Gaussian noise from the original paper (Liang et al., 2021). The denoising from the 50-noise-level model was observed to be inferior to the denoising from the 25-noise-level, and we discarded the 50-noise-level model.

**Evaluation:** Evaluating membrane subtraction requires some care. Recall that the goal of membrane subtraction is to eliminate the membrane signal from the micrograph, but to leave behind any contribution to the micrograph from the membrane protein. In light of this, we evaluate membrane subtraction in two steps. In the first step, we evaluate how membrane-like the estimate of the membrane is for each algorithm. This is done by using the membrane similarity index of Section 3.1 (Equation (2)). As we show below, BM3D and SwinIR have poor membrane similarity indices. Visual examination of their membrane estimates confirm that the estimates do not look membrane-like. Based on this, we eliminate these two methods from further consideration. We then evaluate the remaining two methods (SA and our method) using an index called the *membrane subtraction fraction* which measures the degree to which a membrane is subtracted from the micrograph. This index is based on the empirical observation that the bilayer membrane in a micrograph responds strongly

to a difference-of-gaussians (DoG) filter whose width is approximately the width of the membrane. See Supplementary Text for details. Letting $d_g$ denote the DoG filter kernel, and $I$ and $I_s$ denote the original micrograph and the subtracted micrograph, we define the membrane subtraction fraction as $\eta = \|I_s * d_g\| / \|I * d_g\|$. The smaller this index is, the better the membrane subtraction.

Table 1(a) shows the average membrane similarity index for Data Set 1. BM3D and SwinIR consistently produce poor membrane-like estimates. Further insight into the behavior of these algorithms can be obtained by the example in Figure 4. The top row of this figure shows a vesicle in a Kv1.2 micrograph and the membrane estimates from SwinIR, BM3D, the SA-algorithm, and our algorithm. The bottom row shows the result of subtracting the estimates from the micrograph. Note that the SwinIR membrane estimate has a bead-like appearance and there are several places where noise seems to span the bilayer, or the the bilayer contrast is lost. Note also that the SwinIR subtracted image displays a donut shaped halo where the pixel statistics seem to be different than the noise in the rest of the micrograph. Any membrane protein is unlikely to survive the SwinIR subtraction. The BM3D membrane estimate (top row) does not appear to capture the entire membrane; there is a residual membrane in the subtracted image (bottom row). The SA-algorithm membrane estimate is very membrane-like (top row), but it appears to miss the part of the vesicle near 7 o'clock where the vesicle departs from strict circularity.

| Method | Kv1.2 | KCNQ1 | ATPase |
|---|---|---|---|
| SA-algorithm | **0.85** | **0.84** | **0.81** |
| BM3D | 0.63 | 0.64 | 0.59 |
| SwinIR | 0.60 | 0.76 | 0.55 |
| Our method | 0.78 | 0.78 | 0.77 |

| Method | Kv1.2 | KCNQ1 | ATPase |
|---|---|---|---|
| SA-algorithm | 0.81 | 0.89 | 0.99 |
| Our method | **0.66** | **0.63** | **0.83** |

(a) Membrane similarity index $\nu$ (larger is **better**)  (b) Subtraction Fraction $\eta$ (smaller is **better**)

Table 1: Performance indices for membrane subtraction algorithms

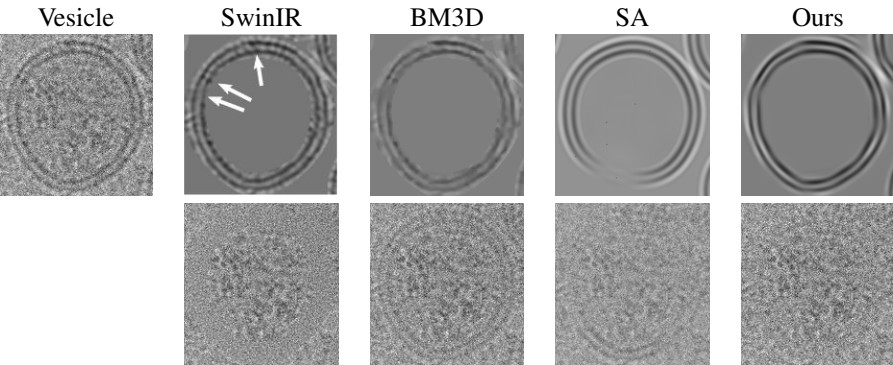

Figure 4: Membrane estimates and membrane subtraction for a Kv1.2 vesicle.

As mentioned above, because of their poor membrane estimates SwinIR and BM3D are discarded from further analysis. Table 1(b) shows the subtraction fraction for the SA-algorithm and our method. The subtraction fraction of our method is significantly superior to that of the SA-method.

Figure 5 illustrates why. The first image in the top row of Figure 5 shows a vesicle from an ATPase micrograph. Below the vesicle is shown the output of the membrane detector (to aid the reader in identifying where the membranes are). The rest of the top row shows membrane estimates from the SA-algorithm and our algorithm. The strong imposition of circularity by the SA algorithm is quite obvious. It is this constraint that results in the poor subtraction fraction for the SA-algorithm. For completeness, the bottom row also shows the result of membrane subtraction.

**Reconstruction:** The ultimate test of successful membrane subtraction is whether the membrane protein can be reconstructed. We used Data set 2 mentioned above to evaluate this. Downsampled (pixel size of $4.3$ Å) micrographs were used with our trained membrane detection model (U-Net)

and our algorithm to detect and estimate the membrane. Downsampling was used to shorten the processing time. The membrane estimate was then upsampled and subtracted from the original micrographs.

After membrane subtraction a geometry-aware particle picker Liu & Sigworth (2014) was used to identify the particles (membrane proteins). This yielded a preliminary set of 180,000 particle images. Subsequent processing and reconstruction used the RELION single-particle pipeline (Scheres, 2012), with some dedicated MATLAB scripts. After a round of 3D classification, the right-side-out particles were selected based on a comparison of the psi (in-plane) angles assigned in RELION and the vesicle-location-based psi angles estimated by the particle picker. This yielded 31,800 particles. Further 3D classification resulted in a final selection of 16,700 particles. Using the same set of particle coordinates, a corresponding set of unsubtracted particle images was also extracted from the original micrographs.



Figure 5: Membrane estimates and membrane subtraction for ATPase vesicles.

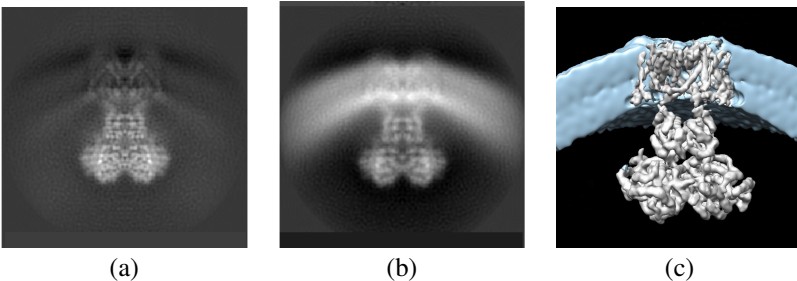

(a)  (b)  (c)

Figure 6: Reconstruction of the Kv1.2 ion channel. (a) 2D projection of the reconstructed Kv1.2 ion channel from membrane-subtracted particle images. (b) 2D projection of the reconstruction from original particle images. (c) 3D rendering of the reconstruction in (b). The membrane protein complex is 150 Å tall.

The RELION 3D auto-refine program was used to create a C4-symmetric membrane-subtracted map with a nominal FSC resolution (Harauz & van Heel, 1986) of 3.7 Å. A 2D projection of this map is shown in Figure 6(a); the transmembrane (TM) part of the protein complex (the upper 1/3 of the complex) shows low density due to the subtraction of the model membrane density, but subtraction also removes almost all signal from the curved membranes. Despite the distortion due to subtraction, the TM region alone has sufficient signal to allow alignment to be performed. The alignment parameters (rotations and translations of the individual particle images) were then used for a 3D reconstruction from unsubtracted particle images. This reconstruction shown in Figure 6(b) contains the true membrane and protein signals, and demonstrates the strong membrane density. Figure 6(c) shows the reconstructed protein and a cutaway view of the membrane density.

## 5 CONCLUSIONS

Here, we report a new unsupervised membrane subtraction algorithm for SPA cryo-EM. Our algorithm depends on a membrane model and optimal local filtering for an estimate of the membrane. This estimate is subtracted from the original micrograph to obtain a membrane-subtracted micrograph. The reported algorithm outperforms previously reported algorithms as well as generic denoising algorithms. Furthermore, the membrane-subtracted micrograph can be used in a standard SPA cryo-EM reconstruction pipeline (i.e., RELION). Reconstruction with real-world images

shows that the transmembrane domain is preserved by the algorithm. Our method facilitates 3D reconstruction of membrane proteins in membranes, better enabling studies of their native states using cryo-EM imaging.

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

# A   APPENDIX

## A.1   DETAILS OF PROPOSITIONS 1 AND 2 OF THE MAIN TEXT

This section of supplementary text contains details of the theory in Section 3 of the main manuscript. Our main goal here is to establish Propositions 1 and 2. We ask the reader to review Section 3.1 before proceeding.

Without any loss of generality suppose that the normal to the membrane at $v_k$ (Figure 7), is in the vertical direction, so that the tangent to the membrane is horizontal. Further assume that the vesicle is circular with radius $R$ (Figure 7). The circle assumption is not as restrictive as it might seem at first glance, because our analysis is local to the vertex $v_k$. Any curve can be locally approximated as a circle.

Fix a point $a$ at a distance $z_2$ above $v_k$ (Figure 7). and consider a horizontal line through the point. The coordinate along the horizontal line is $z_1$. Assume that the true profile function of the membrane is a function of the distance from the center of curvature. Then the value of the true profile function at $a$ is $\zeta(R + z_2)$. The tangentially smoothed estimate of the profile at $a$ is

$$\hat{\zeta}(R + z_2) = \int_{-L}^{L} w(z_1)\zeta(\sqrt{z_1^2 + (R + z_2)^2})dz_1 \tag{13}$$

where $w$ is the weight function.

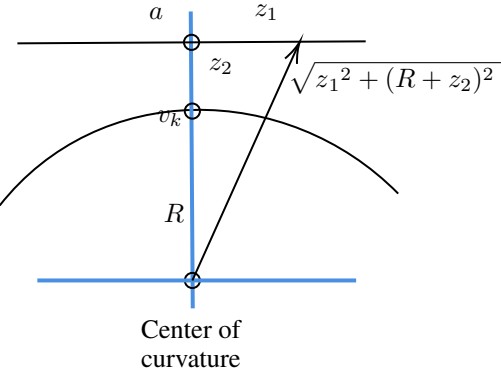

Figure 7: Analysis of tangential smoothing.

What we want to do is to compare the smoothed estimate $\hat{\zeta}(R + z_2)$ with $\zeta(R + z_2)$. The relation between the two is given by the following proposition:

**Proposition S.1:** For $\hat{\zeta}$ and $\zeta$ as defined above:

$$\hat{\zeta}(R + z_2) = \zeta(R + z_2)\int_{-L}^{L} w(z_1)dz_1 + \frac{1}{2}\frac{\zeta'(R + z_2)}{R + z_2}\int_{-L}^{L} w(z_1)z_1^2 dz_1 + O(L^5). \tag{14}$$

**Proof:** For a fixed $z_2$, expanding the true profile $\zeta(\sqrt{z_1^2 + (R + z_2)^2})$ in a Taylor series in $z_1$ around $z_1 = 0$ gives $\zeta(\sqrt{z_1^2 + (R + z_2)^2}) = \zeta(R + z_2) + \frac{1}{2}\frac{\zeta'(R+z_2)}{R+z_2}z_1^2 + O(z_1^4)$. Substituting this series into Equation (13) gives the desired result. ∎

Proposition S.1 suggests something interesting: if we require that the weight function $w$ satisfy $\int_{-L}^{L} w(z_1)dz_1 = 1$ and $\int_{-L}^{L} w(z_1)z_1^2 dz_1 = 0$, then $\hat{\zeta}(R + z_2) \simeq \zeta(R + z_2)$ up to fifth order in $L$. For small $L$ this is a very accurate approximation.

Because $w$ gives a weighted average as $z_1$ varies from $-L$ to $L$, we would additionally like $w$ to be even and smooth. And that it be zero at $-L, L$. This leads to the following calculus of variations

problem:

$$\min_{w} \int_{-L}^{L} w'^2(z_1)dz_1, \quad \text{subject to the following conditions} \tag{15}$$

$$w \text{ is even}, w(L) = w(-L) = 0, \int_{-L}^{L} w(z_1)dz_1 = 1, \int_{-L}^{L} z_1^2 w(z_1)dz_1 = 0.$$

The objective function forces $w$ to be smooth, while the constraints force $w$ to have the properties we want. Note that the above problem is convex in $w$, hence if a solution exists, it is unique. Note also that any solution to this problem depends on $L$. The dependence on $L$ is characterized by the following proposition:

**Proposition S.2:** If $w_1$ is the solution to the above minimization problem for $L = 1$, and $w_L$ is the solution for a general $L$, then $w_L(z_1) = \frac{1}{L}w_1(z_1/L)$.

**Proof:** Suppose $W_L$ is the set of all functions $w$ which satisfy the constraints of the minimization problem in Equation (15) for a given value of $L$. Further for any $w \in W_L$ let $J_L(w)$ be the functional being minimized, i.e. $J_L(w) = \int_{-L}^{L} w'^2(z_1)dz_1$.

Let $\Gamma : W_1 \to W_L$ be a map which takes $f_1 \in W_1$ to $f_L \in W_L$ according to $f_L(z_1) = \frac{1}{L}f_1(z_1/L)$. It is straightforward to show that $\Gamma$ is a bijection. In addition, for any $w_1 \in W_1$

$$J_L(\Gamma(w_1)) = \int_{-L}^{L} (\frac{d}{dz_1}\frac{1}{L}w_1(\frac{z_1}{L}))^2 dz_1 = \int_{-L}^{L} (\frac{1}{L^2}w_1'(\frac{z_1}{L}))^2 dz_1 = \frac{1}{L^3}J_1(w_1).$$

Similarly for any $w_L \in W_L$, $J_L(w_L) = \frac{1}{L^3}J_1(\Gamma^{-1}(w_L))$. That is, under $\Gamma$, the two functionals differ by a fixed scalar multiple. Hence the minimizing functions in $W_1$ and $W_L$ are related by $\Gamma$. ∎

From now on we assume $L = 1$, and for simplicity of notation replace the variable $z_1$ with $x$. For $L = 1$, the solution to the problem of Equation (15) is:

**Proposition S.3:** For $L = 1$, solution to the problem of Equation (15) for $w \in C^2[-1, 1]$ (the space of twice differentiable functions on $[-1, 1]$) is

$$w_1(x) = a_o + a_2 x^2 + a_4 x^4, \quad \text{where } a_0, a_2, a_4 \text{ satisfy}$$

$$a_0 + a_2 + a_4 = 0, a_0 + \frac{1}{3}a_2 + \frac{1}{5}a_4 = \frac{1}{2}, \frac{1}{3}a_0 + \frac{1}{5}a_2 + \frac{1}{7}a_4 = 0. \tag{16}$$

**Proof:** Assuming $L = 1$, ignoring the first two constraints of Equation (15)(we will impose them later), and using Lagrangian multipliers for the equality constraints gives the unconstrained problem:

$$\min_{w} \quad \int_{-1}^{1} w'^2(x)dx + \lambda_1 \int_{-1}^{1} w(x)dx + \lambda_2 \int_{-1}^{1} x^2 w(x)dx \tag{17}$$

The corresponding Euler-Lagrange equation is $w''(x) = \frac{1}{2}\lambda_1 + \frac{1}{2}\lambda_2 x^2$, the solution of which is $w_1(x) = c_0 + c_1 x + \frac{\lambda_1}{4}x^2 + \frac{\lambda_2}{24}x^4$. Requiring that $w_1$ be even gives, $c_1 = 0$. Thus the general form of the function is $w_1(x) = c_0 + \frac{\lambda_1}{4}x^2 + \frac{\lambda_2}{12}x^4$ which can be written as $w(x) = a_0 + a_2 x^2 + a_4 x^4$ ($a_0 = c_0$, $a_2 = \lambda_1/4$, $a_4 = \lambda_2/24$). Imposing the constraints $w_1(1) = 0$, $\int_{-1}^{1} w_1(x)dx = 1$, and $\int_{-1}^{1} x^2 w_1(x)dx = 0$ gives the equations for $a_0, a_2, a_4$ in Equation (16). ∎

Propositions S.1 and S.3 are Propositions 1 and 2 in the main text.

## A.2 EFFECT OF THE DIFFERENCE-OF-GAUSSIAN (DOG) FILTER

This section demonstrates that the output of the DoG filter is sensitive to the presence of the bilayer membrane. Figure 8(a) shows a portion of the micrograph with vesicles (Kv1.2 ion channel). Figure 8(b) shows the output of our algorithm with the membrane subtracted for the same portion of the micrograph. Figure 8(c) and (d) show the output of the DoG filter when applied to Figure 8(a) and (b) respectively (the contrast in Figure 8(c)-(d) has been enhanced to aid visibility). This figure demonstrates clearly that the DoG filter responds strongly to the presence of the membrane.

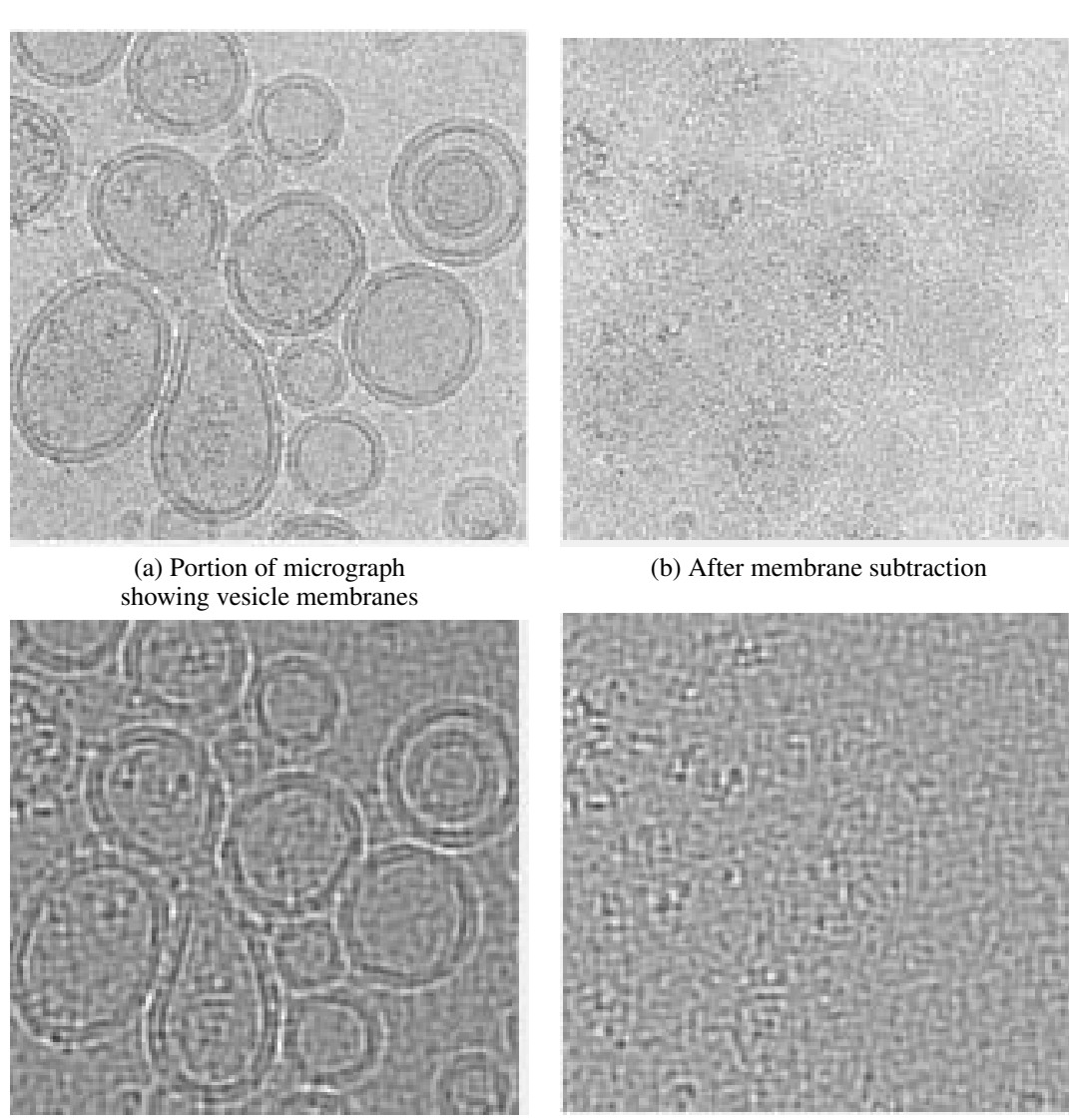

(a) Portion of micrograph showing vesicle membranes

(b) After membrane subtraction

(c) Output of DoG for (a)

(c) Output of DoG for (b)

Figure 8: Response of the DoG filter for micrgraph with vescile membrane and with membrane subtracted.

