# OpenReview forum: "UNSUPERVISED MEMBRANE SUBTRACTION IN CRYOGENIC ELECTRON MICROSCOPY IMAGES"
_ICLR.cc/2026/Conference — Submitted to ICLR 2026_

### Official Review · Reviewer_nL2m · 2025-10-31

**Soundness:** 2
**Presentation:** 3
**Contribution:** 2
**Rating:** 4
**Confidence:** 3

**Summary:**

The paper titled “Unsupervised Membrane Subtraction in Cryogenic Electron Microscopy Images” introduces a new unsupervised framework for subtracting membranes from cryo-EM micrographs, given their segmentation is known. The authors introduce two main novelties. First a mathematical framework for representing biological membranes and second based on that representation an iterative unsupervised algorithm that estimates each unique membrane within micrographs. By estimating and  removing the membrane signal, the method produces micrographs denoised from membrane interference. The proposed framework is evaluated against existing methods, demonstrating better performance in terms of the membrane subtraction fraction. Finally, the authors validate their method’s effectiveness by reconstructing a membrane protein at high resolution.

**Strengths:**

The main strength of the paper lies in its solid mathematical foundation for defining and estimating membrane structures in cryo-EM micrographs. Building on this framework, the authors develop an iterative approach to estimate membrane signal contributions in experimental data and remove them in order to obtain higher-resolution membrane protein reconstructions. The theoretical rigor and clear formulation demonstrate strong technical expertise, along with improved evaluation and presentation. The method has the potential to become a valuable contribution to the field.

**Weaknesses:**

1. The claim of an “unsupervised membrane subtraction algorithm” is not entirely accurate, as the membrane detection step within the pipeline relies on a supervised U-Net model. This dependence weakens the claim of unsupervision.

2. The evaluation is based on limited data diversity, while similar micrographs are used both for training the U-Net and for comparison with other methods. This setup makes the evaluation potentially biased and less fair to the baselines.

3. The experimental analysis could be strengthened through ablation studies (e.g., iteration count, grid spacing, weighting function) and by including more 3D reconstruction results, especially from unseen experiments. Additionally, the validity of the proposed evaluation metrics (the membrane similarity index and subtraction fraction) remains somewhat uncertain, as their correlation with reconstruction quality is not demonstrated.

**Questions:**

1. Please clarify whether the U-Net segmentation output is applied uniformly across all compared methods (SA-algorithm, BM3D, SwinIR) or used only for the proposed approach. This point is essential to assess fairness in the comparison.
2. Could the authors provide evidence or discussion on how the membrane similarity index and subtraction fraction correlate with reconstruction quality? It is unclear whether these metrics meaningfully reflect improvements in 3D reconstruction outcomes.
3. Minor: In line 319 “ pixel sizes of 1.06, 1.09, and 0.825” please add the unit of length.

---

> ### Author Response · Authors · 2025-11-19
>
> Dear reviewer,
>
> Thank you for your valuable input.
> We would first like to address presumed paper weaknesses. Regarding the “unsupervised” claim: there are two stages in the algorithm: (1) membrane detection, followed by (2) membrane subtraction. The “unsupervised” claim only applies to the (2) membrane subtraction. We have made this clear in paragraph 4 of the document, and we will emphasize this point further in the revised manuscript. As we mentioned in the manuscript, membrane segmentation does not present a significant challenge and can be solved by standard supervised segmentation models.
>
> Also, we would like to address the claim that “evaluation is based on limited data diversity”. We believe that our data was quite diverse; it represents different levels of vesicle structure and noise. Almost all vesicles imaged in cryo-EM are either synthetic, reconstructed from natural vesicles, or are natural. The shapes of the vesicles (including radii) differ substantially depending on their type. Furthermore, the signal-to-noise ratio of the micrograph varies depending on imaging conditions such as ice thickness. Our data set spans all of these conditions.  The synthetic vesicles for  K v1.2 (figure 1a) have a limited range of radii and high contrast, the reconstituted vesicles for KCNQ1(figure 1b) have a larger range of radii and lower signal to noise ratio, while the natural vesicles for the ATPase (figure 1c) are significantly non-spherical, have a poor signal to noise ratio, and the “noise” is “asymmetric” with respect to the inside and outside of the membrane. Finally, note that Figure 1 only represents a small snapshot (just 3 micrographs) of vesicle shapes and signal-to-noise ratios in our data. We processed 5,000 micrographs for the reconstruction in Figure 6.
>
> We agree that more 3D reconstructions can bolster our results (as will ablation studies). We hope to carry these out in the immediate future.
>
> Answers to the questions:
>
> 1. Please clarify whether the U-Net segmentation output is applied uniformly across all compared methods (SA-algorithm, BM3D, SwinIR) or used only for the proposed approach. This point is essential to assess fairness in the comparison.
>
> Yes, the U-Net segmentations were used for all methods except one. The exception is the SA-algorithm which uses its own built-in correlation (template matching) instead of segmentation.  For the other algorithms, the denoised images produced by the algorithms were masked by the U-Net segmentation, and the part of the denoised image in the segmentation was used during membrane subtraction.
>
> 2. Could the authors provide evidence or discussion on how the membrane similarity index and subtraction fraction correlate with reconstruction quality? It is unclear whether these metrics meaningfully reflect improvements in 3D reconstruction outcomes.
>
> If the membrane is insufficiently subtracted, then the residual membrane can affect particle reconstruction, especially in the alignment stage of the 3D reconstruction. This is well established in cryo-EM literature. We urge the reviewer to look at Figure 11 d-e in [1] which demonstrates the loss of resolution when the membrane is not subtracted (Figure 11 d) and improved resolution when the membrane is subtracted (Figure 11 e). A discussion of the importance of membrane detection and subtraction is also available in [2] (see second page of the Methods section), where Extended Figure 1 d and e demonstrate the improvement due to membrane subtraction.
>
> The membrane similarity index and subtraction fraction are measures of the ability to subtract membrane. To better understand these measures, consider a vesicle with a protein inserted in the membrane. In the local region of the membrane that contains the protein, the image contours are not likely to be parallel to each other (because of the protein). This loss of parallelism is a signature of the presence of the protein. If this loss of parallelism persists in the membrane estimate, then subtracting the estimate from the micrograph will result in loss of some of the structure of the protein. The membrane similarity index is designed to measure the quality of parallelism in the membrane estimate. The subtraction fraction is a measure of how much of the membrane is subtracted. As mentioned above, [1,2] demonstrates the need to subtract as much of the membrane as possible. The subtraction fraction measures this ability.
>
> References:
>
> [1] K. H. Jensen, S. S. Brandt, H. Shigematsu, F. J. Sigworth, “Statistical modeling and removal oflipid membrane projections for Cryo_EM structure determination of reconstituted membrane proteins”, Journ. Struct. Biol., 194, pp.49-60, 2016.
>
> [2] W. Zheng, P. Chai, J. Zhu, K. Zhang, “High-resolution in situ structures of mammalian respiratory supercomplexes”, Nature, vol. 631, pp 232-239, 4 July  2024.

---

### Official Review · Reviewer_caJJ · 2025-11-01

**Soundness:** 2
**Presentation:** 2
**Contribution:** 2
**Rating:** 4
**Confidence:** 2

**Summary:**

This paper presents an “unsupervised membrane subtraction” approach for cryo-electron microscopy (cryo-EM) images of membrane proteins, aiming to reduce interference from membrane signals during single-particle reconstruction. The method first detects membrane regions using a U-Net model, then performs local geometric modeling with iterative smoothing to estimate and subtract membrane intensity. Experiments on three representative protein systems (Kv1.2, KCNQ1, and Na,K-ATPase) demonstrate that the approach effectively suppresses membrane artifacts while preserving protein details. Overall, the paper proposes a mathematically grounded and reproducible framework that can improve the preprocessing quality of cryo-EM data.

**Strengths:**

The paper tackles a practical problem in cryo-EM image preprocessing with a clear and well-formulated approach.

The method combines geometric modeling and optimization in a coherent framework.

Experimental results clearly show reduced membrane artifacts.

**Weaknesses:**

My background differs somewhat from the authors’, so the following comments are offered from a broader, cross-disciplinary perspective.

Overall, the topic is meaningful and relevant, but from a machine learning and modeling standpoint, the paper’s originality appears moderate. The method mainly systematizes existing geometric and variational techniques rather than introducing new learning or representation mechanisms, and the term “unsupervised” is somewhat misleading given that membrane detection still relies on a supervised U-Net. The experimental dataset is relatively small (around 200 micrographs for training and 5000 for testing in one system) and not publicly available, limiting the assessment of generalization. The comparison includes BM3D, SwinIR, and SA, but omits recent self-supervised or EM-specific baselines. The theoretical derivations are thorough, but the link to implementation details is unclear, and there are minor inconsistencies between text and figures (e.g., missing subpanel in Figure 4).

In summary, the paper demonstrates solid engineering work and numerical stability, yet from a broader ML audience’s viewpoint, it would benefit from stronger methodological originality, more comprehensive experiments, and clearer narrative integration.

**Questions:**

1) Please clarify the scope and assumptions of unsupervised. The paper states supervised membrane detection and “unsupervised” subtraction, but does not formally specify the learning or optimization assumptions for the unsupervised stage (priors, regularizers, observables).

2) Please map theory to implementation explicitly. While implementation details and iterative optimization in PyTorch are described, the correspondence from specific equations (core objectives and regularizers) to actual loss terms and update steps is not clear. An equation to code/loss term mapping would help.

3) Have you considered adding self-supervised or EM-specific baselines? The current comparisons (SA, BM3D, SwinIR) are reasonable but omit domain-relevant unsupervised approaches, which could contextualize the method’s advantages more fairly.

4) Any plans for data release and larger-scale validation? The paper describes two datasets but does not mention public availability or evaluation across more diverse imaging conditions. Releasing sample data or testing on larger datasets would strengthen reproducibility and generality.

BTW, as my expertise is outside the cryo-EM field, these questions are raised from a broader “non-specialist reviewer” perspective. I would appreciate clarifications from the authors on these points and would be interested to see how their responses and the ensuing discussion with other reviewers might address these concerns.

---

> ### Author Response · Authors · 2025-11-20
>
> Dear reviewer,
>
> Thank you for your time, we found your review very insightful.
>
> Below we have some general comments, followed by replies to your questions. Regarding the statement that “experimental data is relatively small and not publicly available”. Many cryo-EM labs do not work on membrane proteins, precisely because membrane subtraction has been an unsolved problem. Nevertheless, given the importance of membrane proteins, there are labs that attempt to reconstruct their structure. We approached three different such labs and obtained micrographs from them. We will approach these labs again to see if data release is possible. It is quite likely that the release of a good membrane subtraction algorithm will lead to many more labs working to membrane proteins.
>
> Also note that almost all vesicles imaged in cryo-EM are either synthetic, or reconstructed from natural vesicles, or are natural. The shapes of the vesicles (including radii) differ substantially depending on the type. Furthermore, the signal to noise ratio of the micrograph varies depending on imaging conditions such as ice thickness. Our data set spans all of these conditions. The synthetic vesicles for K v1.2 (figure 1a) have a limited range of radii and high contrast, the reconstituted vesicles for KCNQ1(figure 1b) have a larger range of radii and lower signal to noise ratio, while the natural vesicles for the ATPase (figure 1c) are significantly non-spherical, have a poor signal to noise ratio, and the “noise” is “asymmetric” with respect to the inside and outside of the membrane. Finally note that Figure 1 only represents a small snapshot (just 3 micrographs) of vesicle shapes and signal-to-noise ratios in our data. We processed 5000 micrographs for the reconstruction in Figure 6.
>
> Regarding the statement that “the link to implementation details is unclear, and there are minor inconsistencies between text and figures (e.g., missing subpanel in Figure 4)”. The implementation is given at the end of Section 3. In brief: fill in the membrane segmentation with a grid. For each grid vertex find $\theta_k$ by minimizing the the right hand side of equation (9) by grid search, with this $\theta_k$ create the basis function of equation (5). Project the data onto the basis functions by minimizing the loss function of equation (7) by gradient descent. Iterate this. We hope to explain this further in the edited version of our paper, if accepted.
>
> Note that Figure 4 is not missing sub panel. Figure 4 consists of five columns, first column is an original image, four following columns are the same fragment but with subtracted membrane by different algorithms. The second row in the columns shows the reconstructed membrane. The first column has this image “missing” because we do not have a ground truth image of just a membrane.
>
> Answers to the questions:
>
> 1. Please clarify the scope and assumptions of unsupervised.
>
> The unsupervised state makes only two assumptions: (1) The membrane model of equation #1, and (2) the linear approximation used in the basis functions in equation #5. The latter is justified in light of Proposition 1 which states that the linear approximation is good to fourth order. There are no priors or regularizers.
>
> 2. Please map theory to implementation explicitly.
>
> Please see our general comments above.
>
> 3. Have you considered adding self-supervised or EM-specific baselines?
>
> Most of the deep learning cryo-EM algorithms that we know of are used for particle picking and reconstruction. The existing denoising algorithms, such as Topaz-Denoise [1], Noise-Transfer2Clean [2] are designed to denoise all the structures in the micrograph, such that to preserve membrane and protein. The heart of the membrane subtraction problem is to estimate the membrane without being influenced by the protein embedded in the membrane. This requires the denoising algorithm to be explicitly aware of the membrane model. We did consider modifying one of the existing algorithms for achieving this, but how a membrane model could be inserted in the existing algorithms was unclear. In part, this was the motivation for developing our algorithm.
>
> 4. Any plans for data release and larger-scale validation? The paper describes two datasets but does not mention public availability or evaluation across more diverse imaging conditions. Releasing sample data or testing on larger datasets would strengthen reproducibility and generality.
>
> As we mentioned above, we obtained data by approaching three different Cryo-EM labs. If this paper is accepted, we will approach the labs for releasing data. So far, one lab has agreed to do so.
>
> References:
>    1. Bepler, Tristan, et al. "Topaz-Denoise: general deep denoising models for cryoEM and cryoET." Nature communications 11.1 (2020): 5208.
>
>  2. Li, Hongjia, et al. "Noise-Transfer2Clean: denoising cryo-EM images based on noise modeling and transfer." Bioinformatics 38.7 (2022): 2022-2029.

---

> > ### Comment · Reviewer_caJJ · 2025-11-26
> >
> > Dear authors,
> >
> > Thanks for the clarifications. It addressed most of my earlier questions, and I really appreciate it. But since I am not from this specific subfield, I would like to wait and see the discussion from other reviewers and the AC.
> >
> > Now I may keep my original score, but I’m open to reconsidering it after hearing their views.

---

### Official Review · Reviewer_yDpV · 2025-11-01

**Soundness:** 1
**Presentation:** 2
**Contribution:** 1
**Rating:** 2
**Confidence:** 4

**Summary:**

The paper addresses the problem of membrane subtraction in cryo-EM images of membrane proteins. To this end, it proposes a mathematical framework to represent membranes in an image and an iterative unsupervised algorithm to subtract them. The paper argues that the proposed method results in better subtraction of membranes than alternate ways.

**Strengths:**

+ A novel mathematical framework is presented for membranes, which to the best of my knowledge, is the first of its kind
+ Experiments were performed on real cryo-EM datasets instead of just simulated data

**Weaknesses:**

- I do not think the problem is significant enough for ICLR conference. Furthermore, the membrane subtraction did not result into significant discoveries, such as, increasing resolution of membrane proteins or discovery of novel membrane proteins.

- The paper itself mentions that proposed mathematical framework for membranes can be used for other membrane-like image features which are not membranes. This is contradicting, in such a case, the definition is wrong. Without taking into account the scale of the image, I do not think it is possible to mathematically define membranes. Membrane-like structures inside the cell are not membranes. Just because the method treats them as so does not make them membranes.

- The reconstruction aspect of the evaluation is not clear. Figure 6 does not demonstrate the benefit of doing membrane subtraction using the proposed method

**Questions:**

Please explain the reconstruction aspect of evaluation. How Figure 6 is showing the need of your membrane subtraction method? Why the alternate methods can not be used?

---

> ### Author Response · Authors · 2025-11-19
>
> Dear reviewer,
>
> We would like to disagree on perceived weaknesses. Perhaps the reviewer is unfamiliar with cryo-EM and its importance in structural biology. Cryo-EM is the modern revolution in structural biology (the inventors of cryo-EM won the Chemistry Nobel Prize in 2017; the NIH has invested $129 million in setting up cryo-EM centers in the US).  Detecting and subtracting membranes is a significant problem in cryo-EM, and in other biological imaging (e.g., cryo-tomography). So far, this problem has been largely unsolved, limiting the use of this approach to only a few laboratories in the world. Our hope is not only to present a solution to this problem, but also to bring it to the attention of the larger machine learning community. One path for machine learning to grow is to look at new problems which have a large user base (most universities and many pharmaceutical companies now have cryo-EM facilities or use one of the national facilities). Membrane subtraction is one such problem. Its solution is likely to have a significant impact on structural biology as many more laboratories will be able to image membrane proteins in their native membrane environment. We also hope that our paper will provide motivation for others in the machine learning community to try their hand at this problem. Cryo-EM is not niche; cryo-EM papers have appeared in ICLR before (e.g., “Reconstructing Continuous Distributions of 3D Protein Structure From Cryo-EM IMAGES” by E. Zhong et al. in ICLR 2020).
>
> We believe that reviewer might misunderstood our claim, we do not claim that our algorithm works with “membrane-like features which are not membranes” but (this is a direct quote from our paper) “can be used with other image features which
> appear membrane-like but are not a lipid bilayer membrane”. Our subtraction algorithm works without change for any membrane, since it only assumes that the membrane has image contours that are parallel to each other. It makes no additional assumptions about the structure of the membrane.
>
> The benefit of membrane subtraction is well-known in cryo-EM: If the membrane signal is insufficiently subtracted, then the residual membrane can affect particle reconstruction, especially in the alignment stage of the 3D reconstruction. This is well established in cryo-EM literature. We urge the reviewer to look at Figure 11 d-e in [1] which demonstrates the loss of resolution when the membrane is not subtracted.  A very recent, pathbreaking paper [2] makes use of partial membrane subtraction done in an ad hoc way. A discussion of the essential nature of membrane detection and subtraction is also given--see the second page of the Methods section, along with Extended Figure 1 d and e in [2].
>
> Answer to the questions:
>
> 1.Please explain the reconstruction aspect of evaluation. How Figure 6 is showing the need of your membrane subtraction method?
>
> The reconstruction uses a standard Cryo-EM package called RELION. RELION is an EM algorithm which reconstructs the 3D structure from projections. The projection angle is treated as a latent variable. Further information on RELION can be found in [3].
>
> The benefit of membrane subtraction is well known in Cryo-EM. We provide references above which clearly demonstrate the advantage of membrane subtraction. But in simple words we could not achieve the same resolution of protein structure without membrane subtraction.
>
>
> 2. Why the alternate methods can not be used?
>
> We clearly showed that alternate methods either do not capture the structure of the membrane (poor membrane similarity index, Table 1 (a)) or do not completely subtract all of the membrane (low subtraction fraction, Table 1 (b)).
>
> References:
>
> [1] K. H. Jensen, S. S. Brandt, H. Shigematsu, F. J. Sigworth, “Statistical modeling and removal of lipid membrane projections for Cryo_EM structure determination of reconstituted membrane proteins”, Journ. Struct. Biol., 194, pp.49-60, 2016.
>
> [2] W. Zheng, P. Chai, J. Zhu, K. Zhang, “High-resolution in situ structures of mammalian respiratory supercomplexes”, Nature, vol. 631, pp 232-239, 4 July  2024.
>
> [3] Scheres SH. RELION: implementation of a Bayesian approach to cryo-EM structure determination. J Struct Biol. 2012 Dec;180(3):519-30. doi: 10.1016/j.jsb.2012.09.006. Epub 2012 Sep 19. PMID: 23000701; PMCID: PMC3690530.

---

> ### Comment · Reviewer_yDpV · 2025-11-19
>
> The reviewer is very well aware of cryo-EM and its importance in structural biology. In fact, the reviewer has regularly published in this domain and is an expert in this field. The reviewer also did not mention cryo-EM is niche, do not blame reviewers for something they did not say.
>
> But the reviewer does think the membrane subtraction is too niche a problem for the ICLR conference. The reviewer disagrees with the response that membrane segmentation is well established in the cryo-EM literature. This is not even relevant for single-particle cryo-EM. Also, the answers the authors provided about the reconstruction do not address my concerns.
>
> The reviewer is well aware of RELION and understands that RELION is used to perform reconstruction. But the reviewer did not understand whether Membrane subtraction by your method + RELION improved on the alternate membrane subtraction method + RELION. The author's response failed to clarify this. If the authors struggle to understand reviewer comments, they are advised to ask clarifying questions rather than blaming reviewers or behaving badly toward them in OpenReview.
>
> Based on the author's response, the reviewer thinks the authors still do not have a clear understanding of the problem background. Hence, the reviewer decided to reduce the score.

---

### Official Review · Reviewer_Lhgf · 2025-11-01

**Soundness:** 2
**Presentation:** 3
**Contribution:** 2
**Rating:** 2
**Confidence:** 5

**Summary:**

Authors propose an image artifact detection method that first estimates the artifact signal by iteratively fitting an artifact-specific mathematical model to the noisy input images. Authors then remove the artifacts by simple subtraction. The scope of the study is limited to membranes in CRYO-EM images.

**Strengths:**

Clarity: The paper is clearly written.
Results: Proposed method outperforms the reported baselines.

**Weaknesses:**

The main issue of the paper is its extremely limited scope. While the work is worthwhile, it is extremely niche to be in the main ICLR conference.


Originality:

Domain specific modelling of objects is a valid approach but iterative smoothing and subtraction steps of the core algorithm is very similar to expectation maximization-based methods like Richardson-Lucy. As far as I can see, the difference here is that the authors' “prior” is their membrane model which is a Gaussian mixture. I do not see this as a significant novelty.


Significance:

The scope of the paper is extremely limited. Authors only address membrane removal in cryo-em. While this is a valid problem to address, I doubt the method is applicable to other tasks (authors do not make such a claim either)  nor is it interesting to the ICLR community.


Experiments:

This is tied to the scope of the paper but results in 3 cryo-em datasets does not tell the reader much.

**Questions:**

Can the proposed smoothing–subtraction procedure be formulated as an optimization problem or probabilistic model to justify the “unsupervised” claim?

---

> ### Author Response · Authors · 2025-11-17
>
> Dear reviewer,
>
> Thank for your time and input.
>
> Please bear with us, we would like to present an alternate point of view about the scope of this paper and the importance of this problem. Cryo-EM is the modern revolution in structural biology (the inventors of cryo-EM won the Nobel Prize in 2013; the NIH has invested $129 million in setting up cryo-EM centers in the US).  Detecting and subtracting membranes is a significant problem in cryo-EM, and in other biological imaging (e.g. cryo-tomography). So far, this problem has been largely unsolved. The three examples we show are from three of the perhaps six labs in the world that are studying membrane proteins in their native membrane environment. Our hope is not only to present a solution to this problem, greatly expanding the usefulness of this cryo-EM approach, but also to bring it to the attention of the larger machine learning community. One path for machine learning to grow is to look at new problems which have a large user base (most universities and many pharmaceutical companies now have cryo-EM facilities or use one of the national facilities). Membrane subtraction is one such problem. Its solution is likely to have a significant impact on structural biology. We also hope that our paper will provide motivation for others in the machine learning community to try their hand at this problem. Cryo-EM in not niche; Cryo-EM papers have appeared in ICLR before (e.g., “Reconstructing Continuous Distributions of 3D Protein Structure From Cryo-EM IMAGES” by E. Zhong et al. in ICLR 2020).
>
> Regarding the importance of membrane subtraction: If the membrane is insufficiently subtracted, then the residual membrane can affect particle reconstruction, especially in the alignment stage of the 3D reconstruction. This effect is well established in the cryo-EM literature. We urge the reviewer to look at Figure 11 d-e in [1] which demonstrates the loss of resolution when the membrane is not subtracted. A very recent, pathbreaking paper [2] makes use of partial membrane subtraction done in an ad hoc way. A discussion of the essential nature of membrane detection and subtraction is given--see the second page of the Methods section, along with Extended Figure 1 d and e in [2].
>
> We also disagree with the comment that our algorithm is similar to Expectation-Maximization (EM). There seems to be some misunderstanding here. The EM algorithm is a maximum-likelihood parameter estimation algorithm which works in the presence of latent variables. Our algorithm does not have any latent variables that are marginalized over, i.e. there is no E-step in our algorithm. Further in our algorithm there are no parameters (of a probability distribution) that are being estimated by maximizing the likelihood, i.e., there is no M-step. Finally, there is no prior in our algorithm (there is a membrane model, but no prior). There is no mention of a Gaussian mixture in our paper.
>
> To be clear, the key idea of our algorithm is that a particular kind of weighted projection on an adapted basis can estimate membrane structure. Further, we provide a theorem which shows how weights can be chosen in an “optimal” way to estimate membranes from noisy data without the knowledge of vesicle curvature.
>
>
> Answer to the questions:
>
> Can the proposed smoothing–subtraction procedure be formulated as an optimization problem or probabilistic model to justify the “unsupervised” claim?
>
> We do not understand this question; this is exactly what we are doing. We have formulated the problem as an optimization problem (find the set of weights which are optimally robust to vesicle curvature and optimally project the data onto a basis defined by such weights).
>
> References:
> [1] K. H. Jensen, S. S. Brandt, H. Shigematsu, F. J. Sigworth, “Statistical modeling and removal oflipid membrane projections for Cryo_EM structure determination of reconstituted membrane proteins”, Journ. Struct. Biol., 194, pp.49-60, 2016.
>
> [2] W. Zheng, P. Chai, J. Zhu, K. Zhang, “High-resolution in situ structures of mammalian respiratory supercomplexes”, Nature, vol. 631, pp 232-239, 4 July  2024.

---

### Official Review · Reviewer_roRs · 2025-11-01

**Soundness:** 3
**Presentation:** 3
**Contribution:** 3
**Rating:** 4
**Confidence:** 4

**Summary:**

This paper proposes an unsupervised method to remove membranes from cryo-EM micrographs while keeping the embedded membrane proteins intact. The goal is to enable 3D reconstruction of membrane proteins without physically extracting them from the lipid bilayer, which often alters their native structure. The method has two main parts: membrane detection (done with a standard supervised U-Net) and membrane subtraction (the new part). The subtraction part models the membrane using local basis functions aligned with membrane curvature, and iteratively smooths and subtracts it. The authors test the method on several datasets of ion channel and ATPase proteins, comparing it to semi-automatic and denoising-based methods. Results show that their approach produces smoother, more realistic membrane estimates and improves 3D reconstructions.

**Strengths:**

* The problem is meaningful for cryo-EM and real biological research.
   * The approach is original and mathematically well grounded.
   * The iterative smoothing framework is intuitive and effective.
   * The results clearly beat older baselines like SA and BM3D.
   * The authors validate their method through both visual results and reconstruction quality.
   * The writing is careful and figures are clear.

**Weaknesses:**

* The unsupervised part still depends on supervised membrane detection.
This makes the claim of “fully unsupervised” less accurate. A discussion on how segmentation errors affect subtraction would help.
   * Besides, the evaluation is quite narrow. Only a few proteins are tested, and all data are from a small number of labs. There is no test on synthetic data or other public cryo-EM datasets. It would be nice to see quantitative results on a broader range of conditions.
   * The mathematical theory section is long and difficult to read. Many parts of Section 3 could be simplified. Readers might find it hard to connect the math to the algorithm.
   * The visual comparison with other methods is convincing but limited.
Adding a quantitative measure of protein preservation would make the results stronger.
   * Finally, computation time is not discussed. The iterative process may be slow on high-resolution images, but this is not reported.

**Questions:**

1. How sensitive is the subtraction to errors in membrane segmentation?
For example, what happens if part of the membrane is missed or mislabeled?
      2. How long does the full process take for a 5760×4092 micrograph? Can it scale to thousands of images efficiently?
      3. Could your algorithm handle multiple membranes or overlapping vesicles in the same image?
      4. Is it possible to use this approach directly on tomograms or 3D cryo-ET data?
      5. What would happen if the membrane is not bilayer-like, for example in irregular cell membranes?
      6. Do you plan to release the code and dataset so that others can reproduce your results?
      7. Could your iterative smoothing be replaced or combined with modern unsupervised denoising networks for speed improvement?

---

> ### Author Response · Authors · 2025-11-19
>
> Dear review,
>
> Thank you for your valuable input.
> First, we would like to address perceived weaknesses. Regarding the “unsupervised” claim: there are two stages in the algorithm: (1) membrane detection, followed by (2) membrane subtraction. The “unsupervised” claim only applies to (2) membrane subtraction. We have made this clear in paragraph 4 of the manuscript, and we will emphasize this point in the revised document.
>
> Regarding “The mathematical theory section is long and difficult to read”: Much of the mathematics in the paper is new (i.e., new to membrane subtraction). The mathematical development is as follows: Section 3.1 defines what the term “membrane” means. Then, equations (3)-(4) define how the membrane is estimated from a noisy image. The details of various operations in (3) and (4) follow. Finally, the algorithm is summarized at the end of Section 3. Because our approach is novel and involves ideas that are not currently prevalent in cryo-EM, we present all mathematical details, including proofs (which are in the supplementary material). We plan to write a full-length journal paper soon, where mathematical development will be more leisurely.
>
> Regarding the comment that “data are from a small number of labs”: very few cryo-EM labs work on membrane proteins in membranes, precisely because membrane subtraction has been a difficult problem. Nevertheless, given the importance of membrane proteins, there are labs that attempt to reconstruct their structure. We approached three different labs and obtained micrographs from them. It is quite likely that the release of a good membrane subtraction algorithm will lead to many more labs working on membrane proteins in a true membrane environment.
>
> Here are the answers to your questions:
>
> 1. How sensitive is the subtraction to errors in membrane segmentation? For example, what happens if part of the membrane is missed or mislabeled?
>
> The main place where membrane detection interacts with membrane subtraction is in determining the vertices v_k of the grid at which the basis functions are located (see Figure 2b in the manuscript). This determination is quite robust to errors in detection. In practice, we observe that subtraction is quite insensitive to changes in membrane detection, as observed, for example, by dilating the detection by several pixels.
>
> 2. How long does the full process take for a 5760×4092 micrograph? Can it scale to thousands of images efficiently?
>
> Micrographs are first downsampled to a pixel size of about 4 Angstroms, which reduces their image size to about 1440x1024 pixels. Before membrane subtraction, the membrane estimate is upsampled, so the original resolution of the micrograph is preserved in the final segmented image. On one NVIDIA RTX 6000 GPU the average segmentation time per micrograph is <1 sec, the subtraction time is <12 sec. We have processed thousands of micrographs successfully, e.g. we processed 5000 micrographs for the reconstruction in Figure 6.
>
> 3. Could your algorithm handle multiple membranes or overlapping vesicles in the same image?
>
> Yes, our algorithm can subtract multiple vesicles and overlapping vesicles in the same image (see vesicles for the Kv1.2 ion channel and KCNQ1 channel in figure 1).
>
> 4. Is it possible to use this approach directly on tomograms or 3D cryo-ET data?
>
> We are working on extending our algorithm for cryo-ET data.
>
> 5. What would happen if the membrane is not bilayer-like, for example in irregular cell membranes?
>
> The membrane model requires the structure to be “membrane like”, meaning that locally it looks like a set of concentric (parallel) contours. There is no requirement for the membrane to be a bi-layer membrane. The subtraction algorithm will work on other membranes without any modification.
>
> 6. Do you plan to release the code and dataset so that others can reproduce your results?
>
> Yes, we will post our code if our paper is published.
>
> 7. Could your iterative smoothing be replaced or combined with modern unsupervised de-noising networks for speed improvement?
>
> We too have wondered about this. The problem is that of informing a generic denoising network of the special structure of membranes (contours are parallel). We are currently working on this.

---

### Meta-Review · Area_Chair_tnn3 · 2026-01-06

**Summary:**

While the reviewers acknowledge the interest of the tackled task and the well-foundedness of the proposed approach, they express concerns regarding the suitability of this work to ICLR. the originality of the method from an ML point of view, the limited scope of the work and evaluation, and the claim of the algorithm being unsupervised.

**Reviewer Concerns:**

The rebuttal convincingly addresses the reviewer's specific questions about the approach. However, the responses provided by the authors regarding the broader concerns of suitability to ICLR and limited scope are unlikely to fully convince the reviewers. As a matter of fact, during the discussion phase, one reviewer mentioned that they would decrease their score, and one that they would maintain it.

**Reviewer Scores:**

Altogether, the reviewers had reached a consensus towards rejection, and it seems unlikely that this would have changed. While there is some interest in the work when it comes to the application domain, a machine learning venue might not be the best fit for it.

---

### Decision · Program_Chairs · 2026-01-26

Reject